# Breaking of Henry's law for sulfide liquid–basaltic melt partitioning of Pt and Pd

Mingdong Zhang [1,2,3] & Yuan Li [1,2✉]

Platinum group elements are invaluable tracers for planetary accretion and differentiation and the formation of PGE sulfide deposits. Previous laboratory determinations of the sulfide liquid–basaltic melt partition coefficients of PGE ($D_{PGE}^{SL/SM}$) yielded values of $10^2$–$10^9$, and values of $>10^5$ have been accepted by the geochemical and cosmochemical society. Here we perform measurements of $D_{Pt,Pd}^{SL/SM}$ at 1 GPa and 1,400 °C, and find that $D_{Pt,Pd}^{SL/SM}$ increase respectively from 3,500 to $3.5 \times 10^5$ and 1,800 to $7 \times 10^5$, as the Pt and Pd concentration in the sulfide liquid increases from 60 to 21,000 ppm and 26 to 7,000 ppm, respectively, implying non-Henrian behavior of the Pt and Pd partitioning. The use of $D_{Pt,Pd}^{SL/SM}$ values of 2,000–6,000 well explains the Pt and Pd systematics of Earth's mantle peridotites and mid-ocean ridge basalts. Our findings suggest that the behavior of PGE needs to be reevaluated when using them to trace planetary magmatic processes.

[1] State Key Laboratory of Isotope Geochemistry, Guangzhou Institute of Geochemistry, Chinese Academy of Sciences, 510640 Guangzhou, China. [2] CAS Center for Excellence in Deep Earth Science, 510640 Guangzhou, China. [3] College of Earth and Planetary Sciences, University of the Chinese Academy of Sciences, 100049 Beijing, China. ✉email: Yuan.Li@gig.ac.cn

Platinum group elements (PGE: Os, Ir, Ru, Rh, Pt, and Pd) are powerful tracers for planetary accretion and core–mantle–crust differentiation[1–5], as well as the formation of magmatic PGE sulfide deposits in Earth's crust[4,6,7]. In planetary magmatic systems, PGE are primarily controlled by their partitioning into sulfide liquid. However, experimental determinations on the partition coefficients of PGE between sulfide liquid and silicate (basaltic) melt ($D_{PGE}^{SL/SM}$) have yielded widely discrepant results, with $D_{PGE}^{SL/SM}$ values ranging from $10^2$ to $10^9$ (refs. [4,8–13]). Studies in the 1990s determined $D_{PGE}^{SL/SM}$ by performing experiments at 1 bar[10–12]. Using a bulk analytical technique, i.e., neutron activation analysis, the obtained $D_{PGE}^{SL/SM}$ for a given element usually vary in the order of $10^2$–$10^6$, which was late ascribed to inefficient separation of sulfide and silicate phases and/or the presence of PGE-sulfide nuggets in the silicate melt[8]. Experimental studies in the 2000s determined $D_{PGE}^{SL/SM}$ by measuring the solubility of PGE in sulfide liquid and silicate melt separately, which usually yielded $D_{PGE}^{SL/SM}$ values in the order of $10^5$–$10^9$ (refs. [14–16]). However, the disadvantage of this "indirect" method is that the formation of PGE-sulfide species in the silicate melt was not considered, which may result in artificially high $D_{PGE}^{SL/SM}$ (refs. [4,8,17]). Most recent experimental studies determined $D_{PGE}^{SL/SM}$ using the in-situ micro-analytical technique of laser ablation–inductively coupled plasma–mass spectroscopy (LA–ICP–MS), such that the contribution of PGE-sulfide nuggets in the silicate melt can be identified and filtered[4,18]. The most precise $D_{PGE}^{SL/SM}$ values obtained at 1200 °C and 1 bar by Mungall and Brenan are larger than $10^5$ (ref. [4]), which have been widely accepted and used by the geochemical and cosmochemical society since their publication in 2014 (refs. [7,8,19,20]). However, in ref. [4], a few wt.% of PGE was doped in the sulfide liquid to facilitate the analysis of PGE in the silicate melt. Whether $D_{PGE}^{SL/SM}$ obey Henry's law when the sulfide liquid has such high PGE concentration has never been investigated. Using the bulk analytical technique, M.E Fleet, J.H Crocket and coauthors first recognized the dependence of $D_{PGE}^{SL/SM}$ on the PGE concentration in the sulfide liquid in the 1990s (refs. [11,12]). However, this has not been further investigated since then, although LA–ICP–MS has been one of the most powerful in-situ micro-analytical techniques from the 2000s. Here we present experimental measurements of $D_{Pt,Pd}^{SL/SM}$ as a function of the Pt and Pd concentration in the sulfide liquid ($C_{Pt,Pd}^{SL}$) using the in-situ analytical technique of LA–ICP–MS. We find that $D_{Pt,Pd}^{SL/SM}$ increase with increasing $C_{Pt,Pd}^{SL}$, and the use of $D_{Pt,Pd}^{SL/SM}$ values of $< 10^4$, which correspond to the $C_{Pt,Pd}^{SL}$ in Earth's magmatic systems, can well explain the observed Pt and Pd systematics of Earth's mantle peridotites and mid-ocean ridge basalts (MORB). These findings suggest that non-Henrian partitioning behavior of PGE should be considered when applying laboratory-determined $D_{PGE}^{SL/SM}$ in planetary magmatic systems.

## Results and discussion

**Pt and Pd partitioning between sulfide liquid and basaltic melt.** Two sets of experiments were performed at 1 GPa and 1400 °C using a piston cylinder apparatus (Supplementary Data 1 and Methods). In Set-1 of 18 forward experiments, ~60–70 wt.% basalt, and ~30–40 wt.% sulfide (FeS) doped with 60–21,000 ppm Pt or 30–7000 ppm Pd, were loaded in graphite capsules. In Set-2 of four reversal experiments, ~200–7000 ppm Pd or 1000 ppm Pt was physically mixed with the basalt and then, ~60–70 wt.% basalt together with ~30–40 wt.% sulfide was loaded in graphite capsules. The formation of PGE nuggets in the silicate melt is related to the oxidation state of the starting silicate[21]. In order to reduce the formation of Pt or Pd nuggets in our basaltic melt, the

starting basalt was prepared to contain only FeO without $Fe_2O_3$. The experimental durations ranged from 24 to 96 h. The experimental oxygen fugacity was close to the C–CO buffer[22–24]. All experiments produced coexisting basaltic melt and sulfide liquid. The basaltic melt was quenched into glass, and the sulfide liquid was quenched into large pyrrhotite crystals and minor Fe-Pt or Fe-Pd alloys, as shown in Fig. 1 and Supplementary Fig. 1. In the quenched sulfide liquid, the proportion of the Fe-Pt or Fe-Pd alloys increases with the mass of Pt or Pd doped in the experiments, and they occur alone the boudary of pyrrhotite crystals. We interpret the Fe-Pt or Fe-Pd alloys as resulting from exsolution of sulfide liquid during quench, as interpreted previouly[4]. We found very small sulfide dots ($< 0.1\,\mu m$) homogeneously distributed in the basaltic glass from all experiments, which we interpret as resulting from quench because of the significant drop in sulfide solubility (Fig. 1 and Supplementary Fig. 1). In a few experiments, we also found that Pt- or Pd-sulfide droplets ($1–5\,\mu m$) dispersed as nuggets/inclusions in part of the basaltic glass, which we interpret as resulting from ineffective segregation from the silicate melt during the run (Supplementary Fig. 1e, f). The basaltic glasses that do not contain macroscopically visible sulfide nuggets ($1–5\,\mu m$) are our analytical targets.

Major and trace element compositions of the quenched basaltic melt and sulfide liquid were measured using electron probe microanalyzer (EPMA) and LA–ICP–MS with large beam sizes (Methods and Supplementary Data 2, 3). The time-resolved LA–ICP–MS signals of Pt and Pd are constant for most of the analyses, confirming the absence of Pt- or Pd-sulfide nuggets in the analyzed basaltic melt from both forward and reversal experiments (Supplementary Fig. 2). Any LA–ICP–MS analyses with Pt or Pd signals contaminated by sulfide nuggets in the silicate melt, as shown in Supplementary Fig. 2g, were discarded. The analytical results show that $C_{Pt,Pd}^{SL}$ equals 60–21,000 ppm and 26–7000 ppm (parts per million by weight), respectively, and the $C_{Pt,Pd}^{SL}$ values obtained using EPMA and LA–ICP–MS agree within 20% relative (Supplementary Fig. 3). $C_{Pt,Pd}^{SM}$ equals 10–160 ppb and 10–120 ppb (parts per billion by weight), respectively, which increases exponentially with increasing $C_{Pt,Pd}^{SL}$ (Fig. 2a, b and Supplementary Data 1). Nernst partition coefficients $D_{Pt,Pd}^{SL/SM}$, as calculated from the weight proportions of Pt and Pd in the sulfide liquid ($C_{Pt,Pd}^{SL}$) and basaltic melt ($C_{Pt,Pd}^{SM}$), range from 1800 to $7 \times 10^5$, which increase linearly with increasing the Pt and Pd concentration in the sulfide liquid (Fig. 2c, d and Supplementary Data 1). No correlation was observed between $D_{Pt,Pd}^{SL/SM}$, the FeO content in the silicate melt, or the atomic metal/S ratio in the sulfide liquid (Supplementary Figs. 4 and 5). At a given Pt and Pd concentration in the sulfide liquid, $C_{Pt,Pd}^{SL}$ and $D_{Pt,Pd}^{SL/SM}$ do not vary as a function of the experimental duration, and the forward and reversal experiments yielded consistent $C_{Pt,Pd}^{SL}$ and $D_{Pt,Pd}^{SL/SM}$ (Fig. 2). These observations suggest the approach of equilibrium partitioning of Pt and Pd. Previous $C_{Pt,Pd}^{SM}$ and $D_{Pt,Pd}^{SL/SM}$, as obtained by Mungall and Brenan using LA–ICP–MS (ref. [4]), are systematically consistent with the present results (Fig. 2).

**Non-Henrian behavior of $D_{Pt,Pd}^{SL/SM}$.** Both the non-linear increase of $C_{Pt,Pd}^{SM}$ and the linear increase of $D_{Pt,Pd}^{SL/SM}$ with increasing $C_{Pt,Pd}^{SL}$ (Fig. 2) indicate that the partitioning of Pt and Pd between sulfide liquid and basaltic melt follows a non-Henrian behavior. Assuming a chalcophile element M dissolves as M-oxide in the silicate melt, the partitioning of M between sulfide liquid and

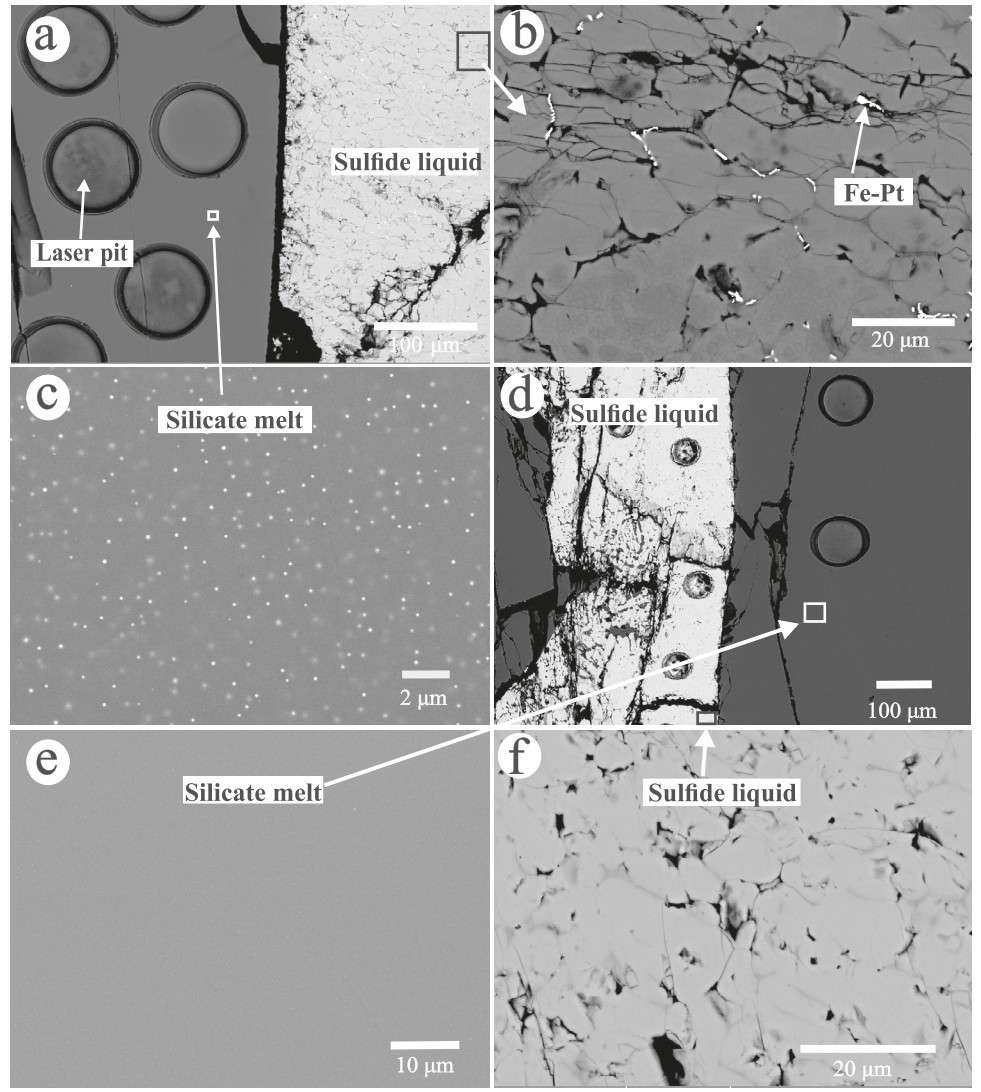

**Fig. 1 Selected back-scattered electron images showing characteristic features of the run products of sulfide liquid–basaltic melt partitioning experiments. a** BSE image of run Z14, showing coexisting sulfide liquid and silicate melt in a graphite capsule. **b** Closer view of the quenched sulfide liquid with 5300 ppm Pt from run Z14, which includes large pyrrhotite crystals and exsolved Fe-Pt alloys at the boundary of pyrrhotite crystals. **c** Closer view of the quenched silicate melt in (**a**), showing homogeneously distributed small sulfide dots (< 0.1 μm) produced during quench. d BSE image of one reversal experiment (run Z28) for Pd partitioning, showing coexisting sulfide liquid and silicate melt. **e** Closer view of the quenched silicate melt in (**d**). **f** Closer view of the quenched sulfide liquid with 163 ppm Pd in (**d**).

silicate melt can be described by the following exchange reaction:

$$\mathrm{MO}_{n/2}(\mathrm{SM}) + n/2\mathrm{FeS}\,(\mathrm{SL}) = \mathrm{MS}_{n/2}(\mathrm{SL}) + n/2\mathrm{FeO}(\mathrm{SM}) \quad (1)$$

where $n$ denotes the valence state of M. The equilibrium constant ($k$) of Eq. (1) can be written as:

$$k_{\mathrm{Eq.(1)}} = \frac{\alpha_{MS_{n/2}}^{SL} \cdot (\alpha_{FeO}^{SM})^{n/2}}{\alpha_{MO_{n/2}}^{SM} \cdot (\alpha_{FeS}^{SL})^{n/2}} \quad (2)$$

and

$$k_{\mathrm{Eq.(1)}} = \frac{x_{MS_{n/2}}^{SL} \cdot \gamma_{MS_{n/2}}^{SL} \cdot (x_{FeO}^{SM})^{n/2} \cdot (\gamma_{FeO}^{SM})^{n/2}}{x_{MO_{n/2}}^{SM} \cdot \gamma_{MO_{n/2}}^{SM} \cdot (x_{FeS}^{SL})^{n/2} \cdot (\gamma_{FeS}^{SL})^{n/2}} \quad (3)$$

In Eqs. (2) and (3) $\alpha$, $x$, and $\gamma$ denote activity, mole fraction, and activity coefficient, respectively, such as $\alpha_{MS_{n/2}}^{SL}$ denoting the activity of $MS_{n/2}$ in the sulfide liquid. Rearranging Eq. (3) would

yield:

$$\frac{x_{MS_{n/2}}^{SL}}{x_{MO_{n/2}}^{SM}} = k_{\mathrm{Eq.(1)}} \cdot \frac{\gamma_{MO_{n/2}}^{SM} \cdot (x_{FeS}^{SL})^{n/2} \cdot (\gamma_{FeS}^{SL})^{n/2}}{\gamma_{MS_{n/2}}^{SL} \cdot (x_{FeO}^{SM})^{n/2} \cdot (\gamma_{FeO}^{SM})^{n/2}} \quad (4)$$

Since $D_M^{SL/SM}$ equals $C \cdot \frac{x_{MS_{n/2}}^{SL}}{x_{MO_{n/2}}^{SM}}$, where $C$ is a constant, the following equation can be obtained:

$$D_M^{SL/SM} = k_{\mathrm{Eq.(1)}} \cdot \frac{\gamma_{MO_{n/2}}^{SM} \cdot (x_{FeS}^{SL})^{n/2} \cdot (\gamma_{FeS}^{SL})^{n/2}}{\gamma_{MS_{n/2}}^{SL} \cdot (x_{FeO}^{SM})^{n/2} \cdot (\gamma_{FeO}^{SM})^{n/2}} \cdot \frac{1}{C} \quad (5)$$

All our experiments were performed at 1 GPa and 1400 °C with very similar silicate melt compositions (Supplementary Data 2); therefore, both $k_{\mathrm{Eq.(1)}}$ and $(x_{FeO}^{SM})^{n/2} \cdot (\gamma_{FeO}^{SM})^{n/2}$ are constant in our experiments. The sulfide liquid contains 60–21,000 ppm Pt or 26–7000 ppm Pd. We therefore assume that $(x_{FeS}^{SL})^{n/2} \cdot (\gamma_{FeS}^{SL})^{n/2}$ is

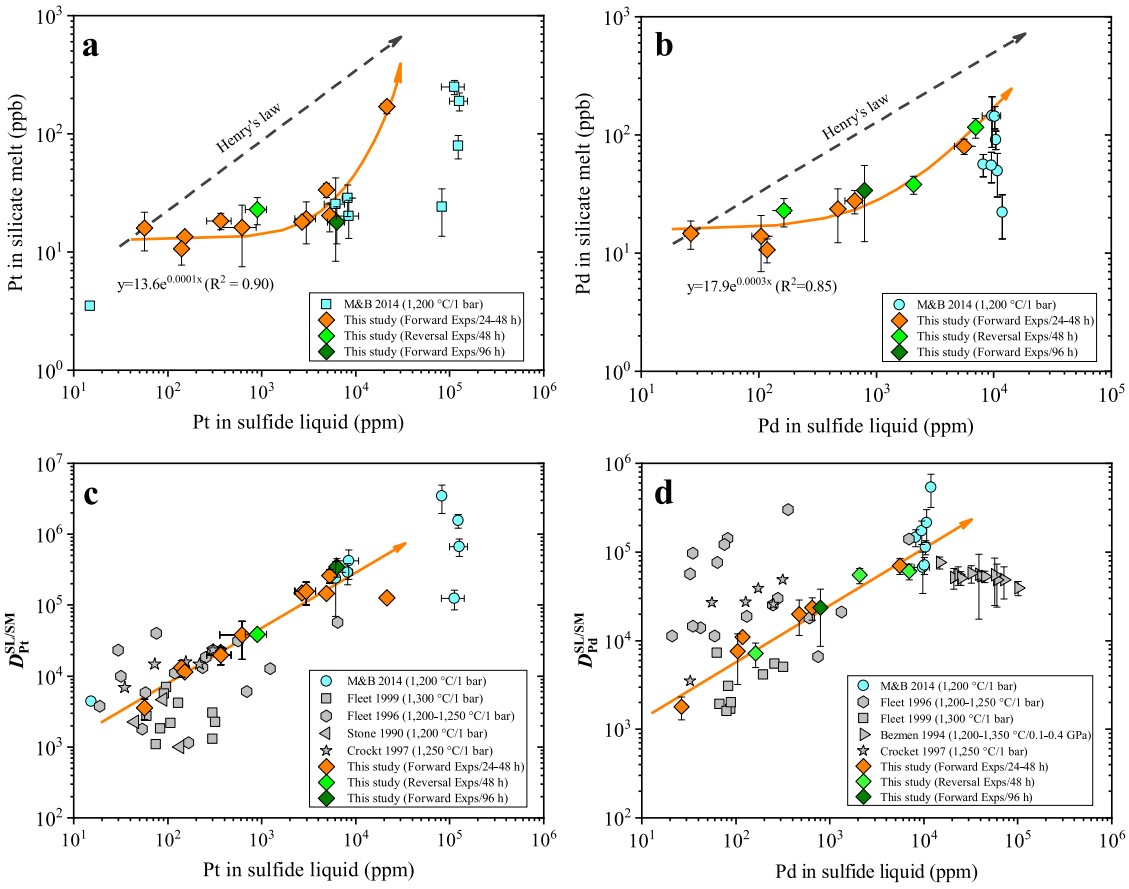

**Fig. 2 Dependence of the Pt and Pd concentration in the silicate melt and $D_{Pt,Pd}^{SL/SM}$ on the Pt and Pd concentration in the sulfide liquid. a, b** The Pt and Pd concentration in the silicate melt increases exponentially with increasing the Pt and Pd concentration in the sulfide liquid. The dashed gray lines show the co-variation of the Pt and Pd concentration in the silicate melt and sulfide liquid if the partitioning of Pt and Pd between sulfide liquid and silicate melt obeys Henry's law. **c, d** $D_{Pt,Pd}^{SL/SM}$ increase linearly with increasing the Pt and Pd concentration in the sulfide liquid. The literature data are taken from M&B 2014 (ref. [4]), Fleet 1996 (ref. [10]), Fleet 1999 (ref. [12]), Stone 1990 (ref. [9]), Bezmen 1994 (ref. [13]), and Crocket 1997 (ref. [11]). Note that the literature data of M&B 2014, obtained using in-situ micro-analytical technique of LA–ICP–MS, are systematically consistent with the present results. The other literature data (filled gray symbols) were obtained by neutron activation analysis of bulk samples, and an effective separation of the silicate and sulfide phases was questioned[8]. Source data are provided in Supplementary Data 1.

also constant in our experiments. Accordingly, the observed increase of $D_{Pt,Pd}^{SL/SM}$ with increasing $C_{Pt,Pd}^{SL}$ can be ascribed to the decrease of the $\gamma_{MS_{n/2}}^{SL}$ for Pt and Pd with increasing $C_{Pt,Pd}^{SL}$, or the increase of the $\gamma_{MO_{n/2}}^{SM}$ for Pt and Pd with increasing $C_{Pt,Pd}^{SM}$. However, when $C_{Pt,Pd}^{SL}$ increases from 60 to 6000 ppm and from 26 to 2000 ppm, respectively, $C_{Pt,Pd}^{SM}$ increases only by a factor of ~2 (Fig. 2a, b), but $D_{Pt,Pd}^{SL/SM}$ increase by a factor of ~30–100 (Fig. 2c, d). Therefore, the $\gamma_{MO_{n/2}}^{SM}$ for Pt and Pd may also be constant, and the increase of $D_{Pt,Pd}^{SL/SM}$ with increasing $C_{Pt,Pd}^{SL}$ most likely reflects that the $\gamma_{MS_{n/2}}^{SL}$ for Pt and Pd decreases with increasing $C_{Pt,Pd}^{SL}$.

We take the advantage of available experimental results[16] on coexisting Fe-Pt alloy and Fe-S-Pt sulfide liquid at 1200–1300 °C and 1 bar to thermodynamically understand the variation of the $\gamma_{MS_{n/2}}^{SL}$ for Pt as a function of $C_{Pt}^{SL}$. Since the activity of Pt in the Fe-Pt alloy can be calculated thermodynamically[16], we can calculate $\frac{1}{k} \cdot \gamma_{PtS_{n/2}}^{SL}$ ($k$ is a constant as shown below) in the Fe-S-Pt sulfide liquid using the following approach. The partitioning of Pt between Fe-S-Pt sulfide liquid and Fe-Pt alloy can be written as:

$$Pt(alloy) + n/4S_2(gas) = PtS_{n/2}(sulfide\,liquid) \qquad (6)$$

where $n$ is the valence state of Pt in the Fe-S-Pt sulfide liquid. The

equilibrium constant of Eq. (6) can be written as:

$$k_{Eq.(6)} = \frac{\alpha_{PtS_{n/2}}^{SL}}{\alpha_{Pt}^{alloy} \cdot (fS_2)^{n/4}} \qquad (7)$$

and

$$k_{Eq.(6)} = \frac{\gamma_{PtS_{n/2}}^{SL} \cdot x_{PtS_{n/2}}^{SL}}{\alpha_{Pt}^{alloy} \cdot (fS_2)^{n/4}} \qquad (8)$$

which can be further arranged as:

$$\frac{1}{k_{Eq.(6)}} \cdot \gamma_{PtS_{n/2}}^{SL} = \frac{\alpha_{Pt}^{alloy} \cdot (fS_2)^{n/4}}{x_{PtS_{n/2}}^{SL}} \qquad (9)$$

In Eqs. (7)–(9) $\alpha_{Pt}^{alloy}$ and $\alpha_{PtS_{n/2}}^{SL}$ are the activity of Pt and $PtS_{n/2}$ in the Fe-Pt alloy and the Fe-S-Pt sulfide liquid, respectively; $\gamma_{PtS_{n/2}}^{SL}$ and $x_{PtS_{n/2}}^{SL}$ are the activity coefficient and mole fraction of $PtS_{n/2}$ in the Fe-S-Pt sulfide liquid, respectively; and $fS_2$ is sulfur fugacity. Since Pt is present mainly as $Pt^{2+}$ in sulfide liquid at geologically relevant redox conditions[16], Eq. (9) can be written as:

$$\frac{1}{k_{Eq.(6)}} \cdot \gamma_{PtS}^{SL} = \frac{\alpha_{Pt}^{alloy} \cdot (fS_2)^{1/2}}{x_{PtS}^{SL}} \qquad (10)$$

Using the experimental results obtained at the same temperatures and similar $fS_2$ values in Fonseca et al.[16], $\frac{1}{k_{Eq.(6)}} \cdot \gamma_{PtS}^{SL}$ was calculated as a function of the Pt concentration in the Fe-S-Pt sulfide liquid (Supplementary Data 4). Figure 3 shows that $\frac{1}{k_{Eq.(6)}} \cdot \gamma_{PtS}^{SL}$ decreases with increasing the Pt concentration in the Fe-S-Pt sulfide liquid. Therefore, $\gamma_{PtS}^{SL}$ must also decrease with increasing the Pt concentration in the Fe-S-Pt sulfide liquid at a given temperature, because $k_{Eq.(6)}$ is a constant at a given temperature. These results thus support our inference that the $\gamma_{MS_{n/2}}^{SL}$ for Pt and Pd decreases with increasing $C_{Pt,Pd}^{SL}$, and our explanations for the non-Henrian behavior of $D_{Pt,Pd}^{SL/SM}$. Here it should be noted that the dissolution of a fraction of Pt and/or Pd as sulfide species in the silicate melt[4,17], which is not considered in Eqs. (1–5), does not change our explanations for the non-Henrian behavior of $D_{Pt,Pd}^{SL/SM}$.

Our experimental results and thermodynamic considerations conclusively demonstrate that $D_{Pt,Pd}^{SL/SM}$ follow a non-Henrian behavior. Therefore, the Pt and Pd concentration must be considered when $D_{Pt,Pd}^{SL/SM}$ are applied in a magmatic system, and the partitioning of Pt and Pd in the sulfide liquid is favored as the

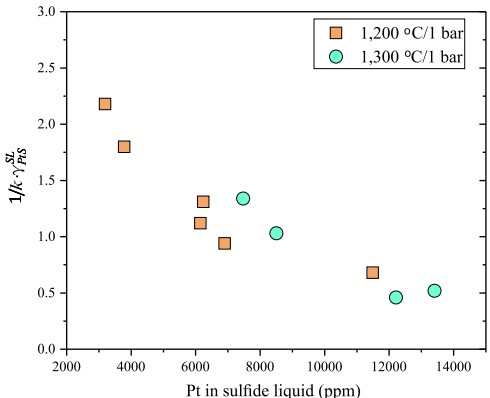

**Fig. 3 The calculated activity coefficient of PtS ($\frac{1}{k} \cdot \gamma_{PtS}^{SL}$) as a function of the Pt concentration in the Fe-S-Pt sulfide liquid.** At the same temperature and pressure and similar $fS_2$, $\frac{1}{k} \cdot \gamma_{PtS}^{SL}$ decreases with increasing the Pt concentration in the Fe-S-Pt sulfide liquid. Note that $k$ is the equilibrium constant of Eq. (6) in the main text. The data used for these calculations were taken from Fonseca et al.[16]. Source data are provided in Supplementary Data 4.

Pt and Pd concentration increases in the magma. The Pt and Pd concentration in sulfides of terrestrial mantle rocks and basalts is usually below 100 ppm[25–27], so the $D_{Pt,Pd}^{SL/SM}$ values of $< 10^4$, rather than the values of $>10^5$ that have been widely accpeted[7,8,20], should be used in terrestrial magmatic systems. We below show that the use of $D_{Pt,Pd}^{SL/SM}$ values of $< 10^4$ well explains the observed Pt and Pd systematics of Earth's mantle peridotites and MORB.

**Model the behavior of Pt and Pd during terrestrial magmatic process.** The newly determined $D_{Pt,Pd}^{SL/SM}$ as a function of $C_{Pt,Pd}^{SL}$ allow us to reconsider the behavior of Pt and Pd during partial melting of Earth's mantle and MORB differentiation. A large dataset of PGE in Earth's mantle peridotites has been established by previous studies[28,29], and considerable variations of the PGE content in Earth's mantle peridotites, as shown in Fig. 4, were explained as resulting from petrogenetic processes such as partial melting and the percolation of mantle fluids/melts[28,30]. To model the Pd content and Pt/Pd ratio in Earth's mantle peridotites (Fig. 4 and Supplementary Data 5), a near-fractional melting model[31] was used. Whole rock $Al_2O_3$ content was taken as a melt depletion indicator, and the melt extraction model parameters for $Al_2O_3$ were taken from ref. [32]. During partial melting, all sulfide was assumed to be sulfide liquid[33]. In addition, the following conditions were used: (i) the starting silicate composition is Earth's primitive mantle[34,35]; (ii) the S concentration at sulfide saturation is 1000 ppm in the generated basaltic melt[36]; (iii) Earth's primitive mantle contains 150–300 ppm S as sulfide (refs. [37,38]); (σ) the $D_{Pt,Pd}^{SL/SM}$ during partial melting of Earth's primitive mantle are 4000 and 2000, respectively; (τ) Pd is completely incompatible in silicate and oxide minerals[5]; (υ) the partition coefficients of Pt between mantle minerals and silicate melt are 0.009 for olivine, 0.8 for clinopyroxene, 2.2 for orthopyroxene, and 0.22 for spinel[39–42]. Figure 4 shows the modeled Pd content and Pd/Pt ratio in Earth's mantle peridotites, which will cover the observed values, particularly if the variations of the S, Pt, and Pd content in Earth's primitive mantle are considered. Accordingly, partial melting of Earth's primitive mantle with $D_{Pt,Pd}^{SL/SM}$ values not exceeding $10^4$ can largely explain the observed Pt and Pd systematics of Earth's mantle peridotites.

We have also modeled the behavior of Pd and Pt during MORB differentiation (Fig. 5 and Supplementary Data 6). The Cu/Pt and Cu/Pd ratios are used as effective tools to identify the timing of sulfide liquid saturation during magmatic differentiation; the Cu/Pt and Cu/Pd ratios in magmas will increase if

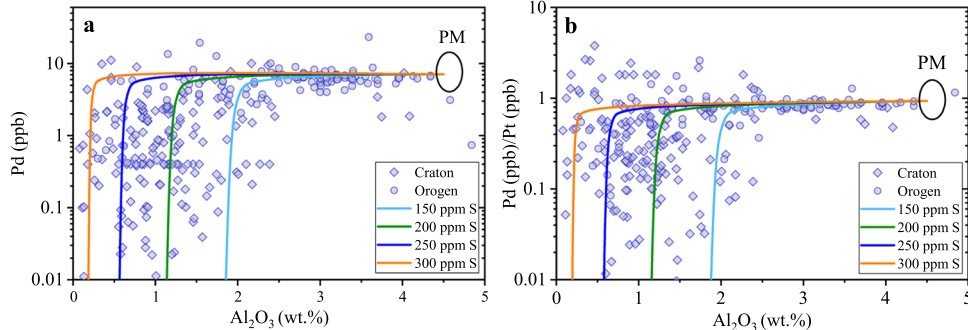

**Fig. 4 Comparison of the Pd content and Pd/Pt ratio of Earth's mantle peridotites with those of our model calculations. a, b** Our model calculations were performed by partial melting of Earth's primitive mantle (PM), with $Al_2O_3$ content (wt.%) in the residual mantle as a proxy for the degree of partial melting. The used $D_{Pt,Pd}^{SL/SM}$ values are 4000 and 2000, respectively. The black ellipses refer to the Pd content and Pd/Pt ratio of the PM (refs. [34, 35]). Mantle peridotite Pt, Pd, and $Al_2O_3$ data were taken from refs. [28, 29] and references therein. This figure illustrates that the Pd content and Pd/Pt ratio in Earth's mantle peridotites can be largely explained by PM partial melting with $D_{Pt,Pd}^{SL/SM}$ values of 4000 and 2000, respectively. Source data are provided in Supplementary Data 5.

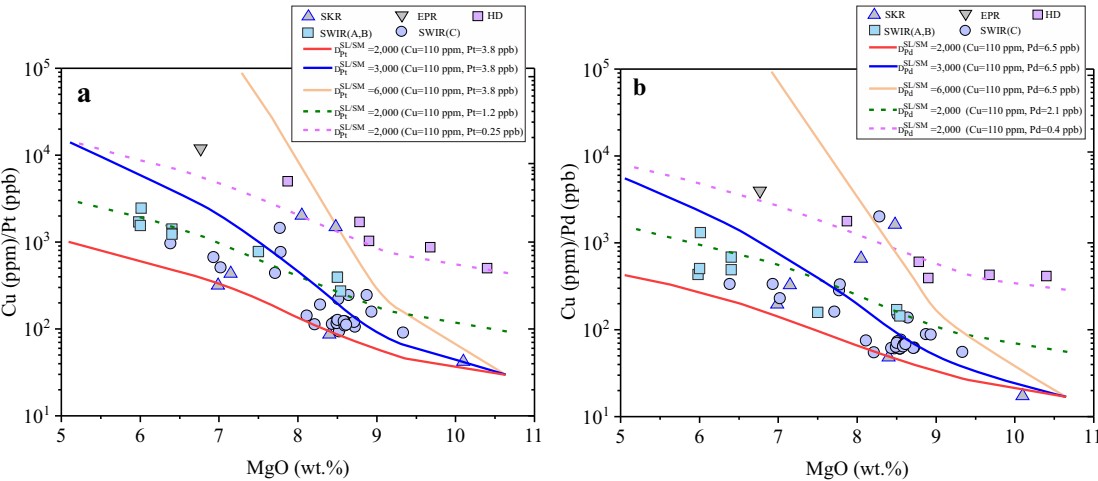

**Fig. 5 Modeled Cu/Pt and Cu/Pd ratios as a function of the MgO content during MORB differentiation. a, b** The $D^{SL/SM}_{Pt,Pd}$ values of 2000–6000 were applied in the modeled differentiation trends, and the observed Cu/Pt and Cu/Pd ratios in MORB can be largely explained. Note that the solid and dashed differentiation lines were modeled using different initial Pt and Pd contents (see main text for details). The MORB Cu, Pt, and Pd data were taken from refs. [5, 43]. SKR Southern Kolbeinsey Ridge, EPR East Pacific Rise; SWIR (A, B): Southwest Indian Ridge Zone A and B; SWIR(C): Southwest Indian Ridge Zone C, HD: HREE-depleted MORBs. Source data are provided in Supplementary Data 6.

sulfide liquid saturation occurs[26,39,43]. To model the evolution trends of Cu/Pt and Cu/Pd during MORB differentiation, we used a fractional crystallization model and assumed that MORB crystallizes isobarically at 0.1 GPa. The relative mass fractions of crystallized minerals (olivine, clinopyroxene, and plagioclase), and the major element composition of the basaltic melt, during MORB differentiation were determined using the Petrolog3 program[44]. We used the SCSS model of Smythe et al.[45] to calculate the S concentration at sulfide saturation in the basaltic melt and the mass of sulfide precipitated, with an assumption that S is removed as sulfide liquid which contains 46 wt.% Fe, 10 wt.% Ni, 7 wt.% Cu, 2 wt.% O, and 35 wt.% S (ref. [26]). Three different groups of Cu, Pt, and Pd content (Cu = 110 ppm, Pt = 3.8 ppb, and Pd = 6.5 ppb; Cu = 110 ppm, Pt = 1.2 ppb, and Pd = 2.1 ppb; Cu = 110 ppm, Pt = 0.25 ppb, and Pd = 0.40 ppb) were used for the parental MORB to differentiate, considering that the parental MORB could have very different Pt and Pd contents[5,43]. The used $D^{SL/SM}_{Pt,Pd}$ values ranged from 2000 to 6000, and the partition coefficients of Pt between minerals and silicate melt are 0.009 for olivine, 0.8 for clinopyroxene, and 0.3 for plagioclase[39,46]. The partition coefficients of Cu between sulfide liquid and basaltic melt were fixed at 800 (refs. [23,24,47]). The partition coefficients of Cu between minerals and basaltic melt are 0.05 for olivine, 0.049 for clinopyroxene, and 0.115 for plagioclase[48,49]. Figure 5 shows that during MORB differentiation, the Cu/Pt and Cu/Pd ratios increase with decreasing the MgO content (wt.%) in the silicate melt, indicating the saturation of sulfide liquid during MORB differentiation[26,50,51]. Figure 5 also shows that the modeled Cu/Pt and Cu/Pd ratios, using $D^{SL/SM}_{Pt,Pd}$ values of 2000–6000, well cover the observed Cu/Pt and Cu/Pd ratios in the differentiated MORB. Therefore, the observed Pt and Pd systematics of MORB can be explained by the use of $D^{SL/SM}_{Pt,Pd}$ values of 2000–6000 for MORB differentiation.

**Implications for the geochemical behavior of PGE in planetary magmas.** The modeling results shown in Figs. 4 and 5 demonstrate that the use of $D^{SL/SM}_{Pt,Pd}$ values of 2000–6000 well explains the observed Pd and Pt systematics of Earth's mantle peridotites and MORB, with $D^{SL/SM}_{Pt,Pd}$ values of >10$^5$ not required. The use of

$D^{SL/SM}_{PGE}$ values of >10$^5$ readily explains the extreme enrichments of PGE in a volumetrically small sulfide fraction and the formation of world-class PGE deposits, such as the Stillwater and Bushveld Complex-hosted reef deposits[6,7,52]. However, our findings, based on both experimental results (Fig. 2) and modeling results (Figs. 4 and 5), suggest that the application of $D^{SL/SM}_{Pt,Pd}$ values of >10$^5$ in terrestrial magmas is inappropriate, and sulfide immiscibility alone cannot explain the observed high Pt and Pd abundance in the Stillwater and Bushveld Complex-hosted reef deposits, which have PGE enrichment factors of 10$^5$–10$^6$ (refs. [6,7,19]). The other mechanisms, such as direct crystallization of PGE-bearing minerals from magmas[53,54], and/or secondary enrichment of PGE in immiscible sulfide liquid[55], may be necessary conditions for the formation of world-class magmatic PGE sulfide deposits.

Our experimental and modeling results also have significant implications for planetary accretion. The $D^{SL/SM}_{Pt,Pd}$ values of <10$^4$ when $C^{SL}_{Pt,Pd}$ <100 ppm suggest that the capacity of segregating sulfide liquid (Hadean matte) in extracting Pt and Pd from Earth's solidifying magma ocean is not as strong as previously thought[3,56]. In this context, the models proposed for Earth's accretion and core-formation based on $D^{SL/SM}_{Pt,Pd}$ values of >10$^5$ (refs. [3,56]) need to be revised to explain the depleted but near-chondritic relative PGE abundance in the bulk silicate Earth. The non-Henrian behavior of $D^{SL/SM}_{Pt,Pd}$ may also suggest that the partitioning of PGE between metallic and silicate melts does not obey Henry's law either. A dependence of the partition coefficients of Pd between metallic and silicate melts (160–3 × 10$^5$) on the Pd concentration in the metallic melt (3000–89,000 ppm) has indeed been observed by Wheeler et al.[57]. Consequently, the distribution of PGE between planetary core and mantle may be controlled by not only temperature and pressure[1–3] but also the PGE concentration in the system. We speculate that it may be possible that the depleted but near-chondritic relative PGE abundance in planetary silicate mantles[58] can be explained by core-formation alone, i.e., a late veneer not being required, if all PGE follow a non-Henrian partitioning behavior between metallic and silicate melts. This speculation deserves future investigations.

Our study conclusively demonstrates that the partitioning of PGE between sulfide liquid and silicate melt is more complex than previously thought. Future laboratory studies must be performed to fully understand the partitioning behavior of PGE between metallic melt, sulfide liquid, and silicate melt, with PGE concentrations close to the natural systems. This will be critically important for tracing many planetary processes, such as the formation of magmatic PGE sulfide deposits in Earth's crust, the origin of the near-chondritic relative PGE abundance in planetary silicate mantles, and the accretion and differentiation of terrestrial planets.

## Methods

**Starting materials**. Starting materials loaded in each sample capsule included ~60–70 wt.% of a synthetic silicate and ~30–40 wt.% sulfide (FeS). The synthetic silicate with major element compositions similar to those of average MORB were prepared from analytical grade oxides and carbonates. To minimize absorbed water, $SiO_2$, $TiO_2$, $Al_2O_3$, $Cr_2O_3$ and MgO powder were each heated overnight at 1000 °C, $MnO_2$ at 400 °C, $CaCO_3$ at 200 °C, and $Na_2CO_3$ and $K_2CO_3$ at 110 °C. After drying, all the oxides and carbonates were mixed and ground in acetone, then dried at room temperature overnight. Well-mixed oxide and carbonate powder was sintered in a high purity alumina crucible at 1000 °C overnight to decarbonate. The decarbonated powder was finally mixed with FeO powder and ground in acetone, so as to limit the introduction of ferric iron into subsequent partitioning experiment, which was proved to be an efficient way to prevent the formation of PGE nuggets in silicate melt[21]. In the forward experiments, ~20–20,000 ppm Pt or Pd was doped in the sulfide powder, which was then sintered in a graphite capsule at 1100 °C and 1.5 GPa for 10 h. In the reversal experiments, ~200–7000 ppm Pd or 1000 ppm Pt was physically mixed with the silicate powder in acetone in an agate mortar, which was then dried at room temperature. All dried silicate and sulfide materials were stored in a vacuumed oven at 110 °C for > 24 h before loading into graphite capsules for high-pressure experiments.

**High temperature and pressure experiments**. All experiments were conducted at 1.0 GPa and 1400 °C in an end-loaded solid media piston cylinder apparatus, using 3/4-inch diameter talc-Pyrex assemblies with graphite heaters. Pressure was calibrated against the quartz-coesite and kyanite-sillimanite transitions, and a friction correction of 18% was applied to the nominal pressure. The total pressure uncertainty is less than 0.1 GPa. The experimental temperatures were monitored by C-type ($W_{95}Re_{05}$-$W_{74}Re_{26}$) thermocouples, and temperature was controlled to ±2 °C and was accurate to ±10 °C. For each experiment, in order to reduce the porosity of the graphite capsules and thus prevent leakage of sulfide liquid, the sample was first heated to 850 °C and held for 2–4 h and then raised to the target temperature. The experimental durations ranged from 24 to 96 h. After quenching by turning off electric power to the graphite heaters, the recovered capsules were sectioned longitudinally into two halves, mounted in epoxy, and polished for EPMA and LA–ICP–MS analyses.

**Electron microprobe analyses of major elements**. Major elements compositions of the quenchend sulfide liquid and silicate melt were measured with a JEOL JXA-8230 microprobe. The analyses were performed in wavelength-dispersive mode, and a PAP matrix correction was applied to the raw data. For the measurements of major elements in the silicate melt, a defocused beam of 20 μm, 15 kV accelerating voltage, and 10 nA beam current were used for both the standardizations and sample measurements. Natural and synthetic standards were used to calibrate the instrument, and the used standards are andradite (Si), $MnTiO_3$ (Ti), spinel (Al), metal Fe (Fe), $MnTiO_3$ (Mn), forsterite (Mg), wollastonite (Ca), albite (Na), orthoclase (K), Gallium phosphite (P), and metal Ni (Ni). The peak counting time was 20 s except for Na and K, which were measured for 10 s. The quenched sulfide liquid was analyzed with 20 kV acceleration voltage and 20 nA beam current. Fe and S were calibrated on a synthetic pyrrhotite with a well-known Fe:S ratio, Ni, Co, Cu, Pd and Pt were calibrated on pure metals, and O was calibrated on magnetite. A defocused beam of 30 μm diameter was used for all standardizations and sample measurements, as this was sufficient to average small-scale quench-phase inhomogeneity. Sulfur in the quenched silicate melts was analyzed with 50 nA beam current and 60 s peak counting time using the synthetic pyrrhotite standard.

**LA-ICP-MS analyses of Pt and Pd**. Major and trace elements of the quenched silicate melt and sulfide liquid were measured using a laser–ablation ICP–MS. These analyses were carried out on an Agilent 7900 Quadrupole ICP-MS coupled to a Photon Machines Analyte HE 193-nm ArF Excimer Laser Ablation system. The quenched silicate melt was analyzed with 10 Hz, 80 mJ, and laser beam sizes of 90–110 μm, whereas the quenched sulfide liquid was analyzed with 7 Hz, 70 mJ, and laser beam sizes of 50–60 μm. A typical time-resolved analysis involved ~20 s of background acquisition, followed by laser ablation for 40 s. The sample chamber

was flushed with He at a rate of 0.4 L/min, to which 5 ml/min $H_2$ was added on the way to the ICP–MS. The isotopes $^{23}$Na, $^{24}$Mg, $^{27}$Al, $^{29}$Si, $^{32}$P, $^{39}$K, $^{42}$Ca, $^{49}$Ti, $^{52}$Cr, $^{55}$Mn, $^{57}$Fe, $^{59}$Co, $^{62}$Ni, $^{65}$Cu, $^{75}$As, $^{121}$Sb, and $^{209}$Bi were measured with a dwell time of 10 ms. To precisely measure $^{105}$Pd and $^{195}$Pt in the silicate melt, $^{105}$Pd and $^{195}$Pt were measured with a dwell time of 50 ms. NIST SRM 610 glass was used as external standard for all analyses, which contains 3.15 ppm Pt and 1.05 ppm Pd[59], whereas Si and Fe determined by electron microprobe were used as the internal standard for the silicate melt and sulfide, respectively. Mass-1 sulfide standard was used to check the accuracy of the LA-ICP-MS analyses. Repeated analyses of the Mass-1 sulfide standard after analyzing each ten sample spots yielded Pt and Pd concentrations consistent with the certified values within 6–17% relative, which demonstrates the validity of using NIST SRM 610 glass as the standard for measuring the Pt and Pd concentration in the quenched sulfide liquid.

As done in previous studies[60,61], the detection limits (DL) for measuring the Pt and Pd concentration ($C_{Pt,Pd}^{SM}$) in the silicate melt were calculated as three times the standard deviation of three replicate measurements of a sample that contains zero analyte[62]. Each analytical point has a own DL because the laser beam size and the interval of signal used for integration may change, althouth a dwell time of 50 ms was always used. The calculated minimum to maximum $DL_{min-max}$ values for the analyses of each sample were given in Supplementary Data 1, which are 3 (min)–7 ppb (max) for Pt and 3 (min)–8 ppb (max) for Pd. Our measured $C_{Pt,Pd}^{SM}$ are significantly higher than $DL_{min-max}$. Most samples have $C_{Pt,Pd}^{SM}/DL_{ave}$ ratios in the range of 3–39 ($DL_{ave}$ = average of DL values for a given sample; Supplementary Data 1). Only five of twenty-two samples show relatively low $C_{Pt,Pd}^{SM}/DL_{ave}$ ratios (2.6–2.9); however, these $C_{Pt,Pd}^{SM}$ values are still well around the quantification limits. The low $C_{Pt,Pd}^{SM}$ (down to ~4 ppb) determined by previous study was confirmed to be reliable under similar analytical conditions[4,21]. In addition, the measured $C_{Pt,Pd}^{SM}$ values of most of our samples have one sigma standard deviation (σ) < 30% relative (Supplementary Data 1). All of these demonstrate the high sensitivity of our LA–ICP–MS analyses of Pt and Pd in the silicate melt and the homogeneous distribution of Pt and Pd in the silicate melt.

## Data availability

All data supporting the findings of this study are available within the paper and supplementary information and data files (Supplementary Data 1–6). Additional data related to this paper may be requested from the authors.

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

## Acknowledgements

We thank Fangyue Wang and Andreas Audétat for helpful discussion on LA–ICP–MS analyses. This work is supported by National Key R&D Program of China (2018YFA0702600) and the National Natural Science Foundation of China (42021002) to Y.L.

## Author contributions

M.D.Z. performed the experiments and analyses as part of his Ph.D. dissertation. Y.L. conceived and supervised the study and interpreted the data. M.D.Z. and Y.L. co-wrote the paper.

## Competing interests

The authors declare no competing interests.
