## [Peer Review File · Nature Communications]

Breaking of Henry's law for sulfide liquid–basaltic melt partitioning of Pt and PdREVIEWER COMMENTS

Reviewer #1 (Remarks to the Author):

Submitted manuscript reports 22 experiments investigating Pt and Pd partitioning between sulfide and silicate liquids. The manuscript concludes that Pt and Pd have non-Henrian behaviour in this system and hence their partition coefficients currently used for modelling terrestrial and planetary processes could be over-estimated.

The study is novel, provocative and relevant to a broad geological audience. It will certainly have a large impact on the future studies of PGE budget in the magmatic systems.

I recommend this manuscript for publication in Nature Communications, subject to relatively minor revisions in the discussion chapter.

Experimental, analytical and thermodynamic parts of this manuscript are very convincing. Some small modifications could be applied to modeling, although most likely they will not have a strong effect on the outcome. Yet, I would recommend the authors to check, in particular, MORB differentiation plotted in figure 4.

I tried to reproduce values for natural Cu/Pt and Cu/Pd ratios and I might be missing something, but I am getting substantially higher values than reported in the plots without even multiplying by 1000. For testing, I used the data by Jenner and O'Neill, 2012 (which I also recommend to use for this plot) and by Yang et al., 2014, referenced by the authors.

Firstly, on the example of Yang et al., 2014, for Cu/Pd ratios I am getting an array between ~60,000 and 1,800,000 (which is huge!) with an average and standard deviation of $259,085 \pm 339,413$. The spread for Cu/Pt ratios is even higher. The number of data points in Yang et al. is 43, while only a small number of points (unclear how they are selected) is plotted in Figure 4. If these are averages, then averages of what? And why isn't standard deviation also plotted? Also, the Y-axis values do not appear right. If I multiply the obtained Cu/Pd ratios by 1000, as suggested according to the Y-axis in Figure 4b, I'll be getting numbers in an order of 10^7 - 10^9 , while the plots shows only values of 10^2 - 10^4 .

Secondly, the rationale for the modelled curves can be somewhat disputed. The authors quote Lee et al. melting model, but for fractional crystallization, Lee et al used constant values of silicate phases that crystallize, and considered only olivine and clinopyroxene. This is in disagreement with a large fraction of plagioclase that usually crystallizes in MORBs. DCu is also unrealistically high for the MORB conditions, even using Li and Audetat values. Li and Audetat report a range of DCu between 540 and 1070 for similar fO_2 s (see figure 3 in Li and Audetat (2012)). If using Kiseeva and Wood (2015) model, then DCu across the liquid line of descent will be roughly varying between ~500 and 800. I do not think that adding plagioclase to the silicates as well as changing DCu by 200-300 would significantly affect the modeling, but it is worth having a look.

Minor comments:

Figure 1C – sulfide blobs are not seen, only when strongly zoomed in.

Figure 3 panel A, typo in the legend, 200 instead of 250 ppm.

References

Jenner, F. E., and O'Neill, H. S., 2012, Analysis of 60 elements in 616 ocean floor basaltic glasses: *Geochemistry Geophysics Geosystems*, v. 13.

Kiseeva, E. S., and Wood, B. J., 2015, The effects of composition and temperature on chalcophile and lithophile element partitioning into magmatic sulphides: *Earth and Planetary Science Letters*, v. 424, p. 280-294.

Li, Y., and Audétat, A., 2012, Partitioning of V, Mn, Co, Ni, Cu, Zn, As, Mo, Ag, Sn, Sb, W, Au, Pb, and Bi between sulfide phases and hydrous basanite melt at upper mantle conditions: *Earth and Planetary Science Letters*, v. 355–356, no. 0, p. 327-340.

Yang, A. Y., Zhou, M.-F., Zhao, T.-P., Deng, X.-G., Qi, L., and Xu, J.-F., 2014, Chalcophile

elemental compositions of MORBs from the ultraslow-spreading Southwest Indian Ridge and controls of lithospheric structure on S-saturated differentiation: *Chemical Geology*, v. 382, p. 1-13.

Reviewer #2 (Remarks to the Author):

The manuscript reports the results of a series of piston-cylinder experiments on the dependence of sulphide melt – silicate melt partition coefficients of Pt and Pd on their respective concentrations in the sulfide liquid. The authors find, that D_{sulfide-silicate} of both Pt and Pd increase with increasing concentration in the sulfide, implying that partitioning does not obey Henry's law, in contrast to what is commonly assumed for partitioning experiments. The topic and findings of the manuscript are interesting and significant to researchers addressing the behaviour of Platinum-Group Elements (PGE) in various processes such as mantle melting or planetary differentiation. However some problems and questions need to be addressed before publication:

- 1) In my opinion, the authors do not sufficiently demonstrate the robustness of their LA-ICP-MS results for Pt and Pd especially at low concentrations in the silicate. How was their detection limit calculated in detail? What is the quantification limit of their measurements, which should actually be used? Demonstrating the accuracy of their measured concentrations is crucial, since the main outcome of the paper depends on correctly determined concentrations. If the low measured concentrations are around or below quantification (!) limit, the Pt and Pd concentration in the silicate may be overestimated, leading to wrongly low partition coefficients.
- 2) What are f_{O2} and f_{S2} (or metal/S ratio) of the experiments? Does the S content of the silicate melt vary with increasing metal in the sulfide? (i.e. decreasing S; e.g. the sulphide of Exp Z25 shows the lowest S content at the highest Pt content). In general even small variations of these parameters exert a main influence on sulphide-silicate partitioning of PGE. So any influence of these parameters on Pt or Pd sulphide-silicate partitioning must be excluded for the presented experiments, before claiming that Henry's law is not obeyed.
- 3) Although it is often used, it doesn't make sense thermodynamically to use wt.%-D's. Instead the authors should use molar D's. (i.e. recalculate wt% concentrations to mole fraction, and from that calculate partition coefficients)
- 4) tables S2 and S3: Some of the EPMA totals are relatively low for silicate glasses and sulfides (e.g. silicates: Z16, Z30, Z25; sulfides: Z18, Z25), Furthermore S dissolves as S²⁻ in silicate melts, replacing oxygen as an anion (e.g. O'Neill & Mavrogenes 2002), so it should not be reported as SO₃, but as S. This would even lead to somewhat lower totals than currently reported.
- 5) When modelling PGE partitioning during planetary accretion, the effect of pressure needs to be taken into account, increasing pressure results in increasing sulphide-silicate partition coefficients. Furthermore, I don't see how the near-chondritic relative abundances of all PGE in the Earth's mantle can be explained by differentiation alone, since the PGE hav. While this might be possible for a single PGE, it is certainly too much of a coincidence that all PGE show chondritic abundances.

Best wishes

Responses to Reviewers' comments

Reviewer #1 (Remarks to the Author):

Submitted manuscript reports 22 experiments investigating Pt and Pd partitioning between sulfide and silicate liquids. The manuscript concludes that Pt and Pd have non-Henrian behaviour in this system and hence their partition coefficients currently used for modelling terrestrial and planetary processes could be over-estimated.

The study is novel, provocative and relevant to a broad geological audience. It will certainly have a large impact on the future studies of PGE budget in the magmatic systems.

I recommend this manuscript for publication in Nature Communications, subject to relatively minor revisions in the discussion chapter.

Experimental, analytical and thermodynamic parts of this manuscript are very convincing. Some small modifications could be applied to modeling, although most likely they will not have a strong effect on the outcome. Yet, I would recommend the authors to check, in particular, MORB differentiation plotted in figure 4.

We thank the Reviewer very much for recognizing the importance and high quality of our work. We have carefully looked at the Reviewer's comments and revised the manuscript accordingly.

I tried to reproduce values for natural Cu/Pt and Cu/Pd ratios and I might be missing something, but I am getting substantially higher values than reported in the plots without even multiplying by 1000. For testing, I used the data by Jenner and O'Neill, 2012 (which I also recommend to use for this plot) and by Yang et al., 2014, referenced by the authors.

Firstly, on the example of Yang et al., 2014, for Cu/Pd ratios I am getting an array between ~60,000 and 1,800,000 (which is huge!) with an average and standard deviation of $259,085 \pm 339,413$. The spread for Cu/Pt ratios is even higher. The number of data points in Yang et al. is 43, while only a small number of points (unclear how they are selected) is plotted in Figure 4. If these are averages, then averages of what? And why isn't standard deviation also plotted? Also, the Y-axis values do not appear right. If I multiply the obtained Cu/Pd ratios by 1000, as suggested according to the Y-axis in Figure 4b, I'll be getting numbers in an order of 10^7 - 10^9 , while the plots shows only values of 10^2 - 10^4 .

Thank you for the very careful comments, which help us correct some mistakes. We have rechecked the data points in Fig.4 (now presented as Fig.5 in the revised

manuscript), we did make a few mistakes.

1. In the last manuscript, we indeed incorrectly labeled the Y-axis, which should be Cu (ppm)/Pt (ppb)/1000, but we labeled it wrongly as Cu (ppm)/Pt (ppb) *1000. We have to apologize for this mistake. To avoid any further confusions, we now directly used Cu (ppm)/Pt or Pd (ppb), without multiplying or being divided by 1000, as the Y-axis.

2. Yes, there are 43 data points in Yang et al., but we missed 7 data points in our last Fig. 4. The missing data points include: (1) five HD-MORB samples; (2) one SKR MORB sample and one SWIR MORB sample. We have now added these data points back to our plots (see our Fig. 5 in the revised manuscript). The great thing is that these data points are well consistent with our modeled curves.

3. Review #1 suggests the data by Jenner and O'Neill, (2012) should be used in our last Fig. 4. We sincerely appreciate this suggestion, but after carefully checking the Pt (but no Pd) data from Jenner and O'Neill, (2012), we found that many MORB Pt data are below the detection limit of their LA-ICP-MS because of the low concentrations of Pt in MORBs. Also, as noted by Jenner et al. (2010) and Jenner and O'Neill. (2012), their Pt data are just approximate contents, which require an interference correction. The Jenner and O'Neill. (2012) MORB trace element dataset has been used widely by the community, but unfortunately, we cannot use the Pt data to compare with our modeled curves at the current stage. But anyway, we appreciate this suggestion.

Secondly, the rationale for the modelled curves can be somewhat disputed. The authors quote Lee et al. melting model, but for fractional crystallization, Lee et al used constant values of silicate phases that crystallize, and considered only olivine and clinopyroxene. This is in disagreement with a large fraction of plagioclase that usually crystallizes in MORBs. DCu is also unrealistically high for the MORB conditions, even using Li and Audetat values. Li and Audetat report a range of DCu between 540 and 1070 for similar fO_2 s (see figure 3 in Li and Audetat (2012)). If using Kiseeva and Wood (2015) model, then DCu across the liquid line of descent will be roughly varying between ~500 and 800. I do not think that adding plagioclase to the silicates as well as changing DCu by 200-300 would significantly affect the modeling, but it is worth having a look.

Thanks for these valuable comments and suggestions. Yes, a large amount of plagioclase occurs during MORB crystallization. In the revised manuscript, plagioclase fraction up to 30% during MORB crystallization was now considered. To determine the relative mass fractions of crystallized minerals, and the basaltic melt composition, during MORB differentiation, we now used the Petrolog3 program. Also, in the revised

manuscript, we used a $D_{\text{Cu}}^{\text{SL/SM}}$ value of 800, according to the experimental work of Li and Audétat (2012) and Kiseeva and Wood (2013, 2015), which value has also been used by others (e.g., Lee et al., 2012). We used a $D_{\text{Pt}}^{\text{plagioclase/silicate melt}}$ value of 0.3 taken from Park et al. (2013).

As the Reviewer realized, adding plagioclase to the crystallized minerals, and changing $D_{\text{Cu}}^{\text{SL/SM}}$ by 200-300, do not affect our model results or conclusions at all (see our Fig. 5 in the revised manuscript, and our Supplementary Table 6).

Jenner, F.E. & O'Neill, H.S.C. Analysis of 60 elements in 616 ocean floor basaltic glasses. *Geochemistry, Geophysics, Geosystems* **13**, n/a-n/a (2012).

Li, Y. & Audétat, A. Partitioning of V, Mn, Co, Ni, Cu, Zn, As, Mo, Ag, Sn, Sb, W, Au, Pb, and Bi between sulfide phases and hydrous basanite melt at upper mantle conditions. *Earth Planet. Sci. Lett.* **355-356**, 327-340 (2012).

Kiseeva, E. S. and B. J. Wood (2013). "A simple model for chalcophile element partitioning between sulphide and silicate liquids with geochemical applications." *Earth and Planetary Science Letters* **383**: 68-81.

Kiseeva, E.S. & Wood, B.J. The effects of composition and temperature on chalcophile and lithophile element partitioning into magmatic sulphides. *Earth Planet. Sci. Lett.* **424**, 280-294 (2015).

Park, J.-W., Campbell, I.H. & Arculus, R.J. Platinum-alloy and sulfur saturation in an arc-related basalt to rhyolite suite: Evidence from the Pual Ridge lavas, the Eastern Manus Basin. *Geochim. Cosmochim. Acta* **101**, 76-95 (2013).

Minor comments:

Figure 1C – sulfide blobs are not seen, only when strongly zoomed in.

We have now taken new back-scattered electron images, and clearly showed the little sulfide blobs quenched from the silicate melts (see revised Fig.1c).

Figure 3 panel A, typo in the legend, 200 instead of 250 ppm.

Thank you for the very careful review. Revised.

References

Jenner, F. E., and O'Neill, H. S., 2012, Analysis of 60 elements in 616 ocean floor basaltic glasses: *Geochemistry Geophysics Geosystems*, v. 13.

Kiseeva, E. S., and Wood, B. J., 2015, The effects of composition and temperature on chalcophile and lithophile element partitioning into magmatic sulphides: *Earth and*

Planetary Science Letters, v. 424, p. 280-294.

Li, Y., and Audétat, A., 2012, Partitioning of V, Mn, Co, Ni, Cu, Zn, As, Mo, Ag, Sn, Sb, W, Au, Pb, and Bi between sulfide phases and hydrous basanite melt at upper mantle conditions: Earth and Planetary Science Letters, v. 355–356, no. 0, p. 327-340.

Yang, A. Y., Zhou, M.-F., Zhao, T.-P., Deng, X.-G., Qi, L., and Xu, J.-F., 2014, Chalcophile elemental compositions of MORBs from the ultraslow-spreading Southwest Indian Ridge and controls of lithospheric structure on S-saturated differentiation: Chemical Geology, v. 382, p. 1-13.

Thank you very much for these recommended references and the largely positive comments on our work.

Reviewer #2 (Remarks to the Author):

The manuscript reports the results of a series of piston-cylinder experiments on the dependence of sulphide melt – silicate melt partition coefficients of Pt and Pd on their respective concentrations in the sulfide liquid. The authors find, that $D_{\text{sulfide-silicate}}$ of both Pt and Pd increase with increasing concentration in the sulfide, implying that partitioning does not obey Henry's law, in contrast to what is commonly assumed for partitioning experiments. The topic and findings of the manuscript are interesting and significant to researchers addressing the behaviour of Platinum-Group Elements (PGE) in various processes such as mantle melting or planetary differentiation. However some problems and questions need to be addressed before publication:

We thank the reviewer very much for the largely positive comments. As can be seen below, we have fully considered and addressed the Reviewer's concerns, which helped us greatly improve the quality of the paper.

1) In my opinion, the authors do not sufficiently demonstrate the robustness of their LA-ICP-MS results for Pt and Pd especially at low concentrations in the silicate. How was their detection limit calculated in detail? What is the quantification limit of their measurements, which should actually be used? Demonstrating the accuracy of their measured concentrations is crucial, since the main outcome of the paper depends on correctly determined concentrations. If the low measured concentrations are around or below quantification (!) limit, the Pt and Pd concentration in the silicate may be overestimated, leading to wrongly low partition coefficients.

We thank the Reviewer very much for these comments, which made us think more deeply about our LA-ICP-MS analyses of Pt and Pd. In our last manuscript, we only briefly mentioned the detection limits (DL) for Pt and Pd. In the revised manuscript, we defined the DL, gave details of the calculation of the DL, and demonstrated the robustness of our LA-ICP-MS analyses of Pt and Pd in the silicate glass.

Similar to previous studies such as Jégo et al. (2010) and Jégo & Pichavant (2012), we calculated DL as three times the standard deviation of three replicate measurements of a sample that contains zero analyte (Longerich et al., 1996). Note that each analytical point has a own DL because the laser beam size and the interval of signal used for integration may change, although a dwell time of 50 ms was always used. Our measured silicate melt Pt and Pd concentrations are significantly higher than DL (3 – 7 ppb for Pt and 3 – 8 ppb for Pd). Most samples have $C_{Pt,Pd}^{SM}/DL_{ave}$ in the range of 3 – 38 ($C_{Pt,Pd}^{SM}$ is the Pt and Pd concentration in the silicate melt). Only a few analyses show relatively low $C_{Pt,Pd}^{SM}/DL_{ave}$ values, down to 2.6; however, these $C_{Pt,Pd}^{SM}$ values are still well around the quantification limits. The low content of Pt and Pd (down to ~4 ppb) in the silicate melt determined by previous study was confirmed to be reliable under similar analytical conditions (Mungall and Brenan, 2014; Médard et al., 2015). In addition, the measured Pt and Pd concentrations in the silicate glass of most of our samples have one sigma standard deviation (σ) less than 30% relative. All of these demonstrate the high sensitivity of our LA-ICP-MS analyses of the Pt and Pd contents in the silicate glass and the homogeneous distribution of Pt and Pd in the silicate glass.

We have added this important information in the Methods section of the revised manuscript, with the following citations. See lines 413 to 429 of the revised manuscript.

Mungall, J.E. & Brenan, J.M. Partitioning of platinum-group elements and Au between sulfide liquid and basalt and the origins of mantle-crust fractionation of the chalcophile elements. *Geochimica et Cosmochimica Acta* **125**, 265-289 (2014).

Médard, E., Schmidt, M.W., Wälle, M., Keller, N.S. & Günther, D. Platinum partitioning between metal and silicate melts: Core formation, late veneer and the nanonuggets issue. *Geochim. Cosmochim. Acta* **162**, 183-201 (2015).

Jégo, S. & Pichavant, M. Gold solubility in arc magmas: Experimental determination of the effect of sulfur at 1000°C and 0.4GPa. *Geochim. Cosmochim. Acta* **84**, 560-592 (2012).

Jégo, S., Pichavant, M. & Mavrogenes, J.A. Controls on gold solubility in arc magmas: An experimental study at 1000°C and 4kbar. *Geochim. Cosmochim. Acta* **74**, 2165-2189 (2010).

Longerich, H.P., Jackson, S.E. and Günther, D. (1996) Inter-laboratory note. Laser ablation inductively coupled plasma mass spectrometric transient signal data acquisition and analyte concentration calculation. *Journal of Analytical Atomic Spectrometry* **11**, 899-904.

2) What are fO_2 and fS_2 (or metal/S ratio) of the experiments? Does the S content of the silicate melt vary with increasing metal in the sulfide? (i.e. decreasing S; e.g. the sulphide of Exp Z25 shows the lowest S content at the highest Pt content). In general even small variations of these parameters exert a main influence on sulphide-silicate partitioning of PGE. So any influence of these parameters on Pt or Pd sulphide-silicate partitioning must be excluded for the presented experiments, before claiming that Henry's law is not obeyed.

All of our experiments were performed at 1.0 GPa and 1,400 °C using a piston cylinder apparatus, and all starting mixtures were loaded into graphite capsules. This approach was used to ensure that the experimental oxygen fugacity was close to the C–CO₂ buffer (Médard et al., 2008; Li and Audetat, 2012; Kiseeva and Wood, 2013).

We also calculated sulfide atomic metal/S ratios, the results show that sulfide metal/S ratios vary within a very small range, and they do not affect the partition coefficients of Pt and Pd (see Figure 1 below). Figure 2 below shows that the S content of the silicate melt does not systematically vary with metal/S ratio in the sulfide. Figure 3 below also show that $D_{Pt,Pd}^{SL/SM}$ do not systematically vary with the FeO content (wt.%) in the silicate melt. Therefore, we can exclude the effects of fO_2 , fS_2 , the FeO content in the silicate melt on $D_{Pt,Pd}^{SL/SM}$.

Figure 1. The S content of the silicate melt as a function of Pt or Pd in the sulfide liquid.

Figure 2. $D_{Pt,Pd}^{SL/SM}$ as a function of the atomic metal/S ratio in the sulfide liquid.

Figure 3. $D_{Pt,Pd}^{SL/SM}$ as a function of the FeO content in the silicate melt.

We have now added this information and figures in the main text, Methods and Supplementary Information of the revised manuscript.

Médard, E., McCammon, C.A., Barr, J.A., and Grove, T.L. (2008) Oxygen fugacity, temperature reproducibility, and H₂O contents of nominally anhydrous piston-cylinder experiments using graphite capsules. *American Mineralogist*, 93, 1838–1844.

Li, Y., and Audétat, A., 2012, Partitioning of V, Mn, Co, Ni, Cu, Zn, As, Mo, Ag, Sn, Sb, W, Au, Pb, and Bi between sulfide phases and hydrous basanite melt at upper mantle conditions: *Earth and Planetary Science Letters*, v. 355–356, no. 0, p. 327-340.

Kiseeva, E.S. and Wood, B.J. (2013) A simple model for chalcophile element partitioning between sulphide and silicate liquids with geochemical applications. *Earth and Planetary Science Letters* 383, 68-81.

3) Although it is often used, it doesn't make sense thermodynamically to use wt.%-D's. Instead the authors should use molar D's. (i.e. recalculate wt% concentrations to mole fraction, and from that calculate partition coefficients)

Thank you for this comment. As realized by the reviewer, wt.-D's (= wt./wt.) are more often used in the geological community. Indeed, wt.-D's are more straightforward and more easily used when using D's to understand magmatic processes in Earth's mantle and crust systems. This is because geoscientists usually report trace element concentrations in minerals and rocks as ppm by wt., rather than as mole fractions.

Following the reviewer's comment, we now also reported molar D's in our Supplementary Table 1 in the revised manuscript for those who may be interested in. However, we still keep the wt.-D's and used wt.-D's to model the behavior of Pt and Pd in magmas.

4) tables S2 and S3: Some of the EPMA totals are relatively low for silicate glasses and sulfides (e.g. silicates: Z16, Z30, Z25; sulfides: Z18, Z25), Furthermore S dissolves as S²⁻ in silicate melts, replacing oxygen as an anion (e.g. O'Neill & Mavrogenes 2002), so it should not be reported as SO₃, but as S. This would even lead to somewhat lower totals than currently reported.

Thank you for these careful comments and suggestions. Now, we have repolished our samples that had relatively low EPMA totals, and we have re-measured their major elements compositions. The new results are presented in Supplementary Tables 2 and 3, with totals between 97 and 99 wt.% for silicates and between 98 and 100 wt.% for sulfides. We believe that the previous relatively poor EPMA totals of a few samples

could be caused by the low polishing quality and rough sample surface. We have also reported the content of S (in ppm) instead of SO₃ in the silicate melt (Supplementary Table 2), as suggested. The deficit of our EPMA totals from 100 wt.% for silicate could be due to the presence of dissolved carbonate and water in the silicate melts, although our experiments are nominally dry.

5) When modelling PGE partitioning during planetary accretion, the effect of pressure needs to be taken into account, increasing pressure results in increasing sulphide-silicate partition coefficients. Furthermore, I don't see how the near-chondritic relative abundances of all PGE in the Earth's mantle can be explained by differentiation alone, since the PGE hav. While this might be possible for a single PGE, it is certainly too much of a coincidence that all PGE show chondritic abundances.

We fully understand the Reviewer's concerns. When modeling PGE partitioning during planetary accretion, the effects of both temperature and pressure should be taken into account, because both parameters may affect the D's of PGE between metallic melt, sulfide liquid, and silicate melt. These have been considered in previous studies focusing on PGE behavior during planetary core formation (e.g., Mann et al., 2012; Rubie et al., 2016; Laurenz et al., 2016; Righter et al., 2008). The community's current understanding is that when temperature and pressure are extrapolated to the conditions of core-mantle segregation in a deep magma ocean, the D's of PGE between metallic and silicate melts are rather small, causing PGE overabundance in Earth's mantle. Therefore, sulfide liquid (Hadean matte) was then proposed to bring additional PGE from the silicate mantle to the core, because the D's of PGE between sulfide liquid and silicate melt were thought to be high during the segregation of sulfide liquid from Earth's mantle.

Since our experimental results show that the partitioning of Pt and Pd between sulfide liquid and silicate melt, and the partitioning of Pd between metallic and silicate melts (Wheeler et al., 2011), do not obey Henry's law, the partitioning of Pt and Pd, and the other PGE by inference, between metallic melt, sulfide liquid, and silicate melt must be strongly controlled by the PGE concentration in the system. Therefore, during planetary core-mantle segregation, and Hadean matte-mantle segregation as well, the distribution of Pt and Pd between planetary core and mantle was controlled by not just temperature and pressure but also the PGE concentration in the system. We speculate that it may be not impossible that the depleted but near-chondritic relative PGE abundance in planetary silicate mantles can be explained by core-formation alone, if the partitioning of all PGE between metallic melt and silicate melt does not follow Henry's law. Of course, future laboratory studies must be performed to fully understand

the partitioning behavior of PGE between metallic melt, sulfide liquid, and silicate melt before we can reach any solid conclusions.

Surely, using non-Henrian law of PGE partitioning to explain the near-chondritic relative abundances of all PGE in the Earth's mantle is just speculation, but it is an important aspect we could consider with our Pt and Pd partitioning data. Please also note that this is not a conclusion of our paper. Rather, it means that we have to re-consider the partitioning behavior of PGE during planetary core formation with the PGE concentrations at the ppb level, close to the natural system, as pointed out in our paper.

We added some key information in the revised manuscript to clarify any potential confusions to the readers. See lines 266 to 271 of the revised manuscript.

Laurenz, V., Rubie, D.C., Frost, D.J. & Vogel, A.K. The importance of sulfur for the behavior of highly-siderophile elements during Earth's differentiation. *Geochim. Cosmochim. Acta* **194**, 123-138 (2016).

Mann, U., Frost, D.J., Rubie, D.C., Becker, H. & Audétat, A. Partitioning of Ru, Rh, Pd, Re, Ir and Pt between liquid metal and silicate at high pressures and high temperatures - Implications for the origin of highly siderophile element concentrations in the Earth's mantle. *Geochim. Cosmochim. Acta* **84**, 593-613 (2012).

Righter, K., Humayun, M. & Danielson, L. Partitioning of palladium at high pressures and temperatures during core formation. *Nat. Geosci.* **1**, 321-323 (2008).

Rubie D C, L.V., Jacobson S A Highly siderophile elements were stripped from Earth's mantle by iron sulfide segregation. *Science* **353(6304)**, 4 (2016).

Wheeler, K.T., Walker, D. & McDonough, W.F. Pd and Ag metal-silicate partitioning applied to Earth differentiation and core-mantle exchange. *Meteorit. Planet. Sci.* **46**, 199-217 (2011).

Best wishes.

We sincerely thank you very much for your great comments, which helped us improve the manuscript significantly!

REVIEWERS' COMMENTS

Reviewer #1 (Remarks to the Author):

I have read the revision of the manuscript and I think the authors have thoroughly addressed all reviewers' comments. I do not have any substantial comments apart from the data presentation in the Supplementary information (see comment below), which can be easily addressed.

This manuscript needs to be proof read by a native-English speaker. A few cosmetic changes are outlined below.

Line 239 – “reconsider” is grammatically incorrect here, unless you would like to imperatively tell the reader to reconsider the geochemical behaviour of PGEs.

Figure 5 is still confusing. There are three different lines (solid red, green dashed and pink dashed) described in the legend identically. Please, change description adding the value of the starting composition or in any other way that makes them different in the legend.

Supplementary Information.

Supplementary tables 1-3. Please, add a number of analyses from which averages and standard deviations were calculated

Supplementary figure 1. This is not critical, but usually an arrow points to the item, not to the text.

Figure captions – lines 19-21, 28, no need for a dash between sulfide and nuggets

Lines 35-36 – Change the font size

Responses to Reviewers' comments

I have read the revision of the manuscript and I think the authors have thoroughly addressed all reviewers' comments. I do not have any substantial comments apart from the data presentation in the Supplementary information (see comment below), which can be easily addressed. This manuscript needs to be proof read by a native-English speaker. A few cosmetic changes are outlined below.

We thank the reviewer very much for the comments, which further help us improve the quality of the paper. We asked a colleague who is native English speaker to read through our manuscript and he helped to spot a few grammatic errors.

Line 239 – “reconsider” is grammatically incorrect here, unless you would like to imperatively tell the reader to reconsider the geochemical behaviour of PGEs.

We replace “reconsider” with “Implications for”. We also change “reconsider” to be “reevaluate” in the abstract part.

Figure 5 is still confusing. There are three different lines (solid red, green dashed and pink dashed) described in the legend identically. Please, change description adding the value of the starting composition or in any other way that makes they different in the legend.

We have revised the Figure legend as suggested (see revised Fig. 5).

Supplementary Information.

Supplementary tables 1-3. Please, add a number of analyses from which averages and standard deviations were calculated

We have added the number of analyses in supplementary Data 1-3.

Supplementary figure 1. This is not critical, but usually an arrow points to the item, not to the text.

Revised (see revised supplementary Fig 1).

Figure captions – lines 19-21, 28, no need for a dash between sulfide and nuggets

Revised.

Lines 35-36 – Change the font size

Revised.